# Distribution of Hematologic Parameters of Complete Blood Count in Anemic and Nonanemic Children in a Mining-Exposed Highland Peruvian Community

**DOI:** 10.3390/ijerph22111637

**Published:** 2025-10-27

**Authors:** Gloria Cruz-Gonzales, Arístides Hurtado-Concha, Héctor Bejarano-Benites, Hernán Bedoya-Vílchez, Merly Sarabia-Tarrillo, Eliane A. Goicochea-Palomino, Jeel Moya-Salazar

**Affiliations:** 1Faculty of Medical Technologist, Universidad Nacional Federico Villarreal, Lima 15007, Peru; ahurtado@unfv.edu.pe (A.H.-C.); hbejarano@unfv.edu.pe (H.B.-B.); habv12052000@gmail.com (H.B.-V.); merlyjazminst23@gmail.com (M.S.-T.); 2Faculties of Health Science, Universidad Tecnológica del Perú, Lima 15046, Peru; elagoichi@gmail.com; 3Faculty of Medicine, Universidad Señor de Sipán, Chiclayo 14002, Peru

**Keywords:** anemia, mining pollution, children, hemoglobin, hematologic tests, pediatrics, Peru

## Abstract

Exposure to heavy metals from mining activities has been consistently associated with disruptions in hematologic homeostasis, adversely affecting children’s overall development. We aimed to determine population-specific distributions of hematological markers and to compare anemic and nonanemic children in a mining-exposed highland community. A cross-sectional study was conducted with 156 children aged 3 to 7 years from the Peruvian highlands, using non-probability sampling and following CLSI C28-A3 guidelines for this population. Inclusion criteria were children with complete blood count results and residency in mining-contaminated areas. Blood samples were collected via venipuncture and analyzed with a 3-part Sysmex differential hematology analyzer. The mean WBC count was 10.42 ± 1.76 × 10^3^/µL, with no significant differences between males and females (*p* = 0.770). Hematological indices, including RBC, hemoglobin, and hematocrit levels, were consistent between sexes. However, significant differences were noted between anemic and nonanemic 3–4-year-old children for RBC (5.56 ± 0.47 vs. 7.06 ± 0.96 × 10^6^/µL) and HCT (33.97 ± 6.89 vs. 35.64 ± 5%) (each *p* < 0.00001), with lower values in anemic subjects. Also, anemic and nonanemic 5–7-year-old children had significant differences in RBC (5.87 ± 1.02 vs. 7.36 ± 0.79 × 10^6^/µL) and HCT (31.13 ± 1.73 vs. 36.54 ± 4) (each *p* < 0.00001). Our findings reveal variations in hematological parameter distributions, emphasizing the importance of personalized blood assessments for mining-exposed populations. This approach could enable earlier diagnosis and intervention for anemia among vulnerable pediatric groups.

## 1. Introduction

Exposure to heavy metals (including lead, mercury, and cadmium) has been consistently linked to disruptions in both immunological and metabolic homeostasis, adversely impacting children’s overall development [1]. International studies [2,3] demonstrate that children in highly polluted areas show significantly elevated concentrations of these metals, which correlate with altered immune cell profiles, such as increased monocytes, eosinophils, neutrophils, and basophils, alongside reduced proportions of natural killer cells [4]. These findings suggest that metal-induced toxicity compromises both innate and adaptive immune responses.

Volatile organic compounds (VOCs), including low-level benzene, are linked to hematological abnormalities. Pediatric studies demonstrate significant disparities in leukocyte, erythrocyte, and platelet counts between children in high-VOC areas versus low-pollution regions [5]. Prenatal exposure to methylmercury and persistent organic pollutants (POPs) is correlated with reduced leukocytes (notably lymphocytes), while perfluoroalkyl substances (PFASs) are associated with elevated basophil counts [6]. These findings indicate that co-exposure to multiple toxic agents modulates immune function through multifactorial mechanisms, potentially initiating hematopoietic disruptions prenatally.

Heavy metal exposure is strongly associated with decreased hemoglobin levels, a critical factor in anemia pathogenesis. The inverse relationship between lead exposure and hemoglobin concentration is well-documented, with even low-level metal exposure significantly impairing hematological status in children [7,8,9]. Similarly, mercury exposure, particularly from artisanal mining, is linked to elevated biological mercury levels and subsequent hemoglobin reduction, directly compromising oxygen transport and exacerbating neurodevelopmental and physical growth deficits [6].

The clinical repercussions of these hematological alterations manifest as a spectrum of symptoms ranging from fatigue, pallor, irritability, learning difficulties, and growth impairment, with potential lifelong consequences for cognitive function and quality of life [10,11]. Thus, mining-related pollutants pose a dual threat, not only as drivers of anemia but also as contributors to systemic health inequities in vulnerable populations [12]. Accumulating epidemiological evidence underscores the urgent need for targeted surveillance and prevention strategies to mitigate environmental contamination’s impact on children, particularly in regions with intensive mining activity [13]. Systematic monitoring of hematological markers adapted to high-exposure settings is essential for prompt diagnosis, timely intervention, and informed public health policies [14,15].

Common hematological parameters (i.e., hemoglobin, hematocrit) under mining conditions may be influenced by the toxicological loads of Andean highland communities exposed to mining contaminants or by the effects of malnutrition in these communities [16,17]. While limited studies have characterized hematological changes in anemic children within these settings [18], establishing baseline distributions of normal and abnormal parameters remains a critical first step in defining the hematological landscape of this vulnerable population.

In view of these challenges, we aimed to determine the distribution of hematological markers in anemia-affected and non-anemic children from Peru’s mining-exposed highlands.

## 2. Materials and Methods

### 2.1. Study Design and Settings

This cross-sectional study enrolled children from Pacucha, a mining-exposed community in Peru’s Apurímac Andes. The region hosts active formal and artisanal mining, with ≥9 copper/zinc extraction sites [19], linked to environmental heavy metal contamination that disproportionately endangers underserved populations with limited healthcare access [20,21,22]. Evidence indicates that the most polluting metals from mining in Apurimac are Zn, Cr, Cu, Pb, As, and Ni [23,24]. Anemia imposes a disproportionately high burden here, affecting 61.9% of children, a rate escalating markedly over the past decade [19,25].

### 2.2. Sampling, Inclusion Criteria, and Variables

Non-probabilistic sampling was used in accordance with Clinical and Laboratory Standards Institute (CLSI) C28-A3 guidelines [26], yielding a sample size of ≥120 participants. Inclusion criteria were children aged 3 to 7 years, of either sex, with available complete blood count (CBC) results, and residing in areas exposed to mining-related contamination. Exclusion criteria included foreign nationals, as they are often nomadic in Peru and therefore may live for a few months in the mining area and then move, which could add bias to the study by preventing true assessment of changes related to the environment. Also, children who were hospitalized were also excluded, as they were not available in the mining area. They were instead referred to the regional hospital, which is not accessible.

Primary variables included 14 hematological parameters and anemia. Anemia was defined according to Ministry of Health (MINSA) of Peru [27]. For children aged 24 to 59 months, mild anemia is defined as having hemoglobin levels between 10 and 10.9 g/dL, moderate anemia as 7 to 9.9 g/dL, and severe anemia < 7 g/dL. For those aged 5 to 11 years, mild anemia is indicated by hemoglobin levels from 11 to 11.4 g/dL, moderate anemia from 8 to 10.9 g/dL, and severe anemia at levels < 8 g/dL. Additional variables such as sex (male/female), age [3–4 (n = 74) and 5–7 years (n = 84) for analysis, according to the MINSA anemia guideline] [27], and geographic origin were also recorded.

### 2.3. Instruments and Serological Markers

A 3-part Symex XP-300 differential hematology analyzer (Sysmex, Kōbe, Japan) was used to determine total white blood cell count (WBC, 10^3^/µL), granulocyte count (GRA% and GRA#), lymphocyte count (LYM% and LYM#), and mid-sized white cells (MID% and MID#, including monocytes, eosinophils, and basophils). Red blood cell count (RBC, 10^6^/µL), hemoglobin concentration (HGB, g/dL), hematocrit (HCT, %), mean corpuscular volume (MCV, fL), mean corpuscular hemoglobin (MCH, g/dL), mean corpuscular hemoglobin concentration (MCHC, pg/dL), and platelet count (PLT, 10^3^/µL) were also measured. The automated counter was calibrated twice by the manufacturer before the study began, and three daily control levels were utilized throughout the analysis of the children’s samples. The analysis took place at the Lira Laboratory in the city of Andahuaylas. The Laboratory participates in an external quality program provided by the manufacturer with performance evaluations every semester as previous reported [28].

### 2.4. Data Gathering and Analysis

Between January and March 2023, community health representatives, along with the mining company’s occupational physicians, contacted parents or guardians of prospective participants by phone during an annual population screening. Blood samples were collected via venipuncture using lavender-top BD Vacutainer tubes. Samples were processed within three hours of collection according to standardized laboratory protocols. Two authors (A.H-C., and M.S-T.) were responsible for sample collection and storage. The handling, storage (4 °C) and transportation of samples followed previous international and national recommendations [29,30,31,32], and samples were processed within 4 h of morning sampling [33]. Data were exported directly from the analyzer’s system to a Microsoft Excel 2019 database.

Descriptive statistics (mean ± standard deviation, simple frequencies) were computed. Percentile values were estimated using the Tukey and Dixon-Reed methods [34], and lower (2.5th percentile) and upper (97.5th percentile) limits were stratified by sex, age and anemia status. All analyses were conducted using DATAtab version 1.0 (Graz, Austria) and SPSS v26.0 (IBM Statistics, Armonk, NY, USA).

### 2.5. Ethical Aspects

This study was conducted in accordance with the principles of the Declaration of Helsinki [35] and was approved by the Ethics Committee of the Universidad Nacional Federico Villarreal (Protocol No. 104-2021). Written informed consent was obtained from parents or guardians, and assent was obtained from participating children. All data were anonymized and managed in compliance with Peruvian regulations [36].

## 3. Results

A total of 156 children were included, with a mean age of 4.79 ± 1.46 years (range: 3–7), of whom 82 (52.6%) were female. Most participants were from the district of Chiara (21.8%) and Huayana (19.2%), followed by Huancarama (17.3%), Talavera (15.4%), and Huancaray (12.2%) (Table 1). The overall prevalence of anemia in the children evaluated was 30.8%, while the prevalence of anemia in children aged 3–4 and 5–7 years was 18.9% and 41.5%, respectively.

The mean total WBC count was 10.42 ± 1.76 × 10^3^/µL (range: 6.6–15.2). Differential counts were 4.05 ± 1.44 cells/mm^3^ (range 1.7–7.4), 0.78 ± 0.33 cells/mm^3^ (range 0.3–1.6) y 5.57 ± 1.52 cells/mm^3^ (range 2.6–11.5) for LYM, LYM, MID and GRA, respectively. Corresponding percentages were LYM% were 38.5 ± 11.02% (range 18.7–60.7), MID% were 7.34 ± 2.87% (range 0.30–15.80), and GRA% were 54.06 ± 12.6% (range 31–77.8), respectively. For red blood cell parameters, RBC were 6.98 ± 1.08 × 10^6^/µL (range 0.92–9.35), HGB were 12.32 ± 2.01 g/dL (range 8.7–22), and HCT were 35.35 ± 4.71% (range 25.2–53). Hematimetric indices were 50.50 ± 2.85 fL (range 42.5–65.4), 17.33 ± 1.06 g/dL (range 15.8–65.4 gr/dL), and 34.31 ± 1.11 pg/dL (range 32.5–40.1) for MCV, MCH, MCHC, respectively. The PLT was 226.47 ± 35.1 × 10^3^/µL (Figure 1).

### 3.1. Hematological Parameters Distribution by Sex

There were no significant differences between the sexes in leukocyte counts and their subsets, expressed as absolute numbers or percentages (Table 2). Correspondingly, red blood cell indices and platelet counts were comparable between males and females. The analysis of leukocyte parameters revealed a distinct profile. The total WBC count was elevated at 10.42 ± 1.76 × 10^3^/µL, while a differential count analysis demonstrated a relative and absolute lymphocytosis, with LYM# of 4.05 ± 1.44 × 10^3^/µL and a mean LYM% of 38.50 ± 11.02%. Correspondingly, the GRA% was 54.06 ± 12.60%. No statistically significant differences were observed in any leukocyte parameter between male and female participants (all *p*-values > 0.05). Platelet counts (PLT) were within normal reference ranges and did not differ by sex.

### 3.2. Hematological Parameter Distribution According Age Groups

Overall, most parameters showed no statistically significant differences between the two groups (3–4- and 5–7-year age groups), suggesting relative stability of hematologic indices across this developmental range. Mean total WBC counts were similar in both groups (10.28 ± 1.74 vs. 10.55 ± 1.78 × 10^3^/μL; *p* = 0.350), as were absolute lymphocyte (3.94 ± 1.45 vs. 4.15 ± 1.43 × 10^3^/μL; *p* = 0.362) and granulocyte counts (5.59 ± 1.43 vs. 5.56 ± 1.61 × 10^3^/μL; *p* = 0.906). Notably, the only parameter that reached statistical significance was the percentage of MID cells, which was higher in the 5–7-year-old cohort (7.85 ± 2.93%) compared to 3–4-year-olds (6.79 ± 2.72%; *p* = 0.021) (Table 3).

While WBC counts were similarly elevated in both the 3–4-year and 5–7-year age groups (10.28 ± 1.74 vs. 10.55 ± 1.78 × 10^3^/µL, *p* = 0.350), a significant age-dependent increase was observed in the proportion of MID%, which rose from 6.79 ± 2.72% in the younger children to 7.85 ± 2.93% in the older group (*p* = 0.021). The absolute count of MID cells showed a non-significant trend toward increase (0.73 ± 0.31 vs. 0.83 ± 0.34 × 10^3^/µL, *p* = 0.053). No other leukocyte subset or platelet count demonstrated statistically significant variation with age.

RBC counts, HGB levels, and HCT also showed no significant intergroup differences (*p*-values of 0.369, 0.085, and 0.474, respectively). MCV, MCH, and MCHC showed comparable values between the groups (*p* > 0.05 in each case), indicating that basal erythrocyte indices remained consistent across these age groups. Platelet counts were also stable (226.61 ± 31 vs. 226.34 ± 38.6 × 10^3^/μL; *p* = 0.962), indicating that there were no major thrombocyte variants in this population.

### 3.3. Hematological Distribution According Anemia Outcomes

Significant differences were observed between the anemic and nonanemic groups for key parameters such as RBC, HGB levels, and HCT (*p* < 0.00001 for all comparisons), with lower values consistently observed in the anemic subgroups across both age ranges (Figure 2).

Assessment of erythrocyte indices revealed age-dependent variations. In the 3–4-year-old cohort, the MCV did not differ significantly between anemic and nonanemic children (*p* = 0.223). In contrast, within the 5–7-year-old group, anemic children exhibited a significantly higher MCV (*p* = 0.0018). Similarly, the MCH demonstrated significant differences between groups in both age categories (*p* = 0.001 for 3–4 years and *p* = 0.0004 for 5–7 years), with more pronounced variations observed in the older cohort. The MCHC was significantly different between anemic and nonanemic children only in the 3–4-year age group (*p* = 0.020), whereas no statistically significant difference was detected in the 5–7-year-old group (*p* = 0.165) (Table 4).

## 4. Discussion

Our results demonstrated that, while most distributions of hematological parameters were similar across sexes and age groups, children tested at all ages with anemia exhibited significantly lower RBC and HCT values. In addition, the prevalence of anemia affected a third of children in Peruvian mining settings.

### 4.1. Strengths

A key strength of this study lies in its focus on a narrow age range, a critical developmental window for pediatric neurological and physiological maturation and for detecting early hematological effects of heavy metal exposure [37,38,39]. Unlike prior studies that analyze broader age groups [40,41] or focus only on infants [37,39] or adolescents [40], our targeted approach minimizes confounding variability linked to developmental stages. Furthermore, while most research prioritizes RBC- and HGB-related parameters due to their direct association with anemia [37,38,39,41,42], our inclusion of WBC and platelet counts enabled identification of additional heavy metal-induced hematopoietic effects beyond anemia [43]. This study is also, to the best of the authors’ knowledge, among the first to estimate hematological distribution in mining-exposed Peruvian children with and without anemia.

### 4.2. Main Findings

Although age and sex are known to dynamically influence pediatric hematological normal ranges from infancy through adolescence [44,45,46], our analysis revealed no significant sex- or age-based differences in distribution; however, anemia status dramatically altered erythrocyte parameters. This study revealed the distribution of red blood cell parameters in a population of children in a mining environment. This is important to understand how these parameters behave in anemia, as they can be affected by altitude and pollution conditions from mining activities.

In mining communities, children face concurrent exposure to multiple toxic metals, whose synergistic interactions may amplify adverse health effects, particularly on hematological parameters [40]. When anemia coexists, it creates a dual burden, as mean impaired oxygen transport from anemia compounds the direct toxic effects of heavy metals, worsening clinical outcomes. While mercury and cadmium exposure has shown sporadic associations with hematological alterations [37,38], the strongest and most consistent evidence links lead exposure to iron-deficiency anemia in children. For instance, blood lead levels ≥5 µg/dL are frequently associated with reduced erythrocyte parameters and heightened anemia risk, potentially exacerbating susceptibility to lead absorption and perpetuating a cycle of toxicity [39,41]. Our findings align with this evidence, as anemia significantly correlated with lower RBC, HGB, and HCT values and differentially influenced erythrocyte indices across age groups.

Notably, our inclusion of WBC and platelet analyses, rarely prioritized in heavy metal–anemia studies [37,38,39,41,42], uncovered potential hematopoietic disruptions. The consistently elevated WBC counts across all age groups and both sexes suggest a persistent, population-wide stimulus, most likely chronic inflammation or recurrent subclinical infections. For example, elevated WBC counts in lead-exposed children may reflect subclinical inflammation or bone marrow stress [43], suggesting broader systemic impacts warranting further investigation. This is a common feature in populations with high environmental pathogen exposure and poor sanitation, conditions often exacerbated in mining communities [47].

Multimetal exposure not only drives hematological abnormalities but also precipitates diverse clinical symptoms, including fatigue, pallor, irritability, and, in severe cases, impaired learning, growth, and cognitive function [15,16]. The age-dependent rise in the MID% fraction, primarily comprising monocytes and immature cells, is particularly revealing. This shift may indicate an escalating antigenic burden or chronic immune activation as children grow older and have cumulative exposure to mining-related contaminants like heavy metals or silica dust [48,49]. These agents can act as haptens or cause tissue damage, provoking a sustained innate immune response [50,51]. Further research is needed to identify the effect of these contaminants on leukocytes and their link to anemia in children.

Mitigating these outcomes requires integrated strategies, as environmental controls to reduce metal exposure, nutritional interventions (i.e., iron supplementation), and community education programs targeting caregivers. Additionally, socioeconomic factors (parental education, occupation, household income, and housing conditions) significantly influence exposure risks and must inform policy design [40,42,52]. Regular updates and validation of hematological data in high-exposure settings are equally critical to improve diagnostic accuracy, guide timely interventions, and optimize public health planning [38,40,42].

### 4.3. Limitations

First, we did not assess exposure routes (i.e., environmental, dietary, breast milk) or duration (i.e., transient vs. lifelong residency in mining areas), which are critical for understanding cumulative metal interactions and their hematological impacts [40]. Second, while we adhered to CLSI guidelines for the estimation of hematological distribution, reference interval assessment is required incorporating expanded age groups and diverse mining-exposed communities. Key gaps include the absence of anemia subtype markers (i.e., total iron-binding capacity, ferritin) and verification of normal reference intervals using manual laboratory methods (i.e., microhematocrit, smear-based cell counts, HGB estimation formulas) [53,54]. Third, while the Sysmex XP-300 analyzer proved reliable in this study, its performance must be benchmarked against widely used hematology systems in Peru (e.g., GENRUI KT-40, DYMIND DH36, Landwind LW D3600) through rigorous quality control protocols [55,56]. Fourth, in this study, we did not analyze the frequency, type of exposure to mining toxins, or the concentration of common toxins, so the distribution of hematological parameters could be influenced by these factors and amend the interpretation of the results. Another limitation was the absence of supplementary data needed to interpret anemia types. While identifying the causes of anemia was not the study’s goal, hematology reports should incorporate peripheral blood smear analysis, other parameters like RDW (indicating iron deficiency and B12/folate deficiency), and assessment alongside leukocyte metrics, such as WBC in inflammation cases, or detailed analysis of MID%. This should be considered in future research. Finally, unmeasured socioeconomic factors (parental education, income, housing quality) and nutritional status may confound exposure pathways and health outcomes, necessitating their inclusion in future studies [42,52].

## 5. Conclusions

This study establishes the first hematological distribution values in children from mining regions of Peru, revealing that, in the state of anemia, significant reductions in red blood cell count and hematocrit were found, regardless of age. The results of this research may be useful in understanding the distribution of hematological values in children aged 3 to 7 years in mining areas, thus promoting future research that will improve healthcare and child protection policies for vulnerable children in Peru.

## Figures and Tables

**Figure 1 ijerph-22-01637-f001:**
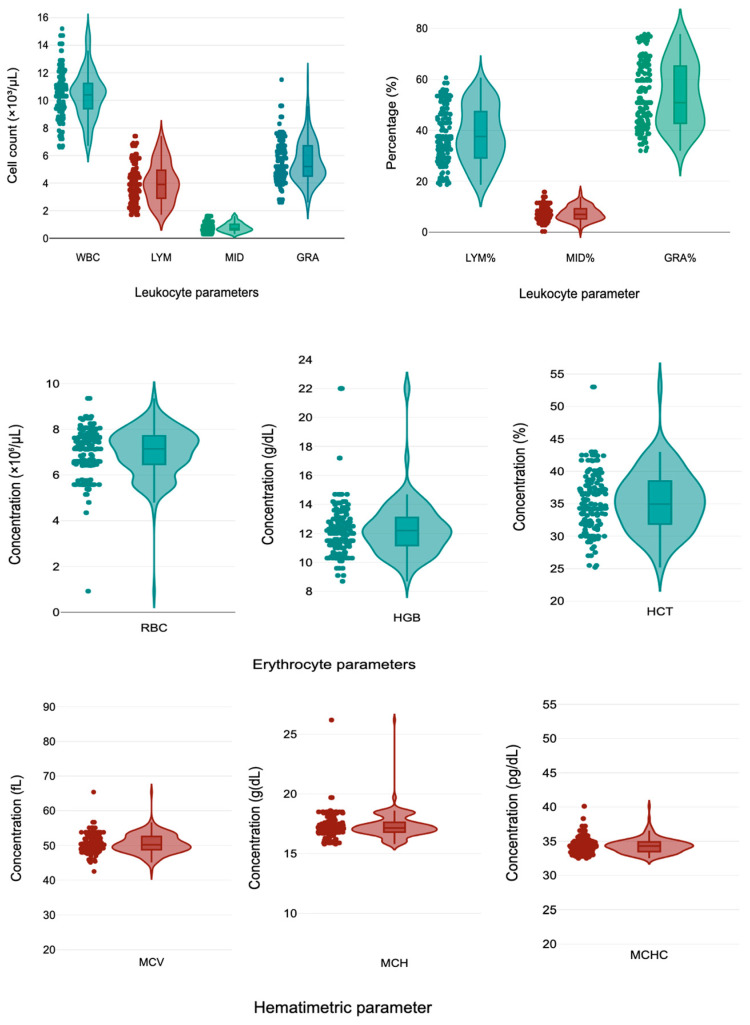
Distribution of hematological parameters in children in a mining-exposed highland Peruvian community. Abbreviations: WBC: white blood cell count, GRA% and GRA#: granulocyte count, LYM% and LYM#: lymphocyte count, MID% and MID#: mid-sized white cells, RBC: Red blood cell count, HGB: hemoglobin concentration, HCT: hematocrit, MCV: mean corpuscular volume, MCH: mean corpuscular hemoglobin, MCHC: mean corpuscular hemoglobin concentration. The colored violin plots only indicate the distribution of hematological parameters, and the data points are shown.

**Figure 2 ijerph-22-01637-f002:**
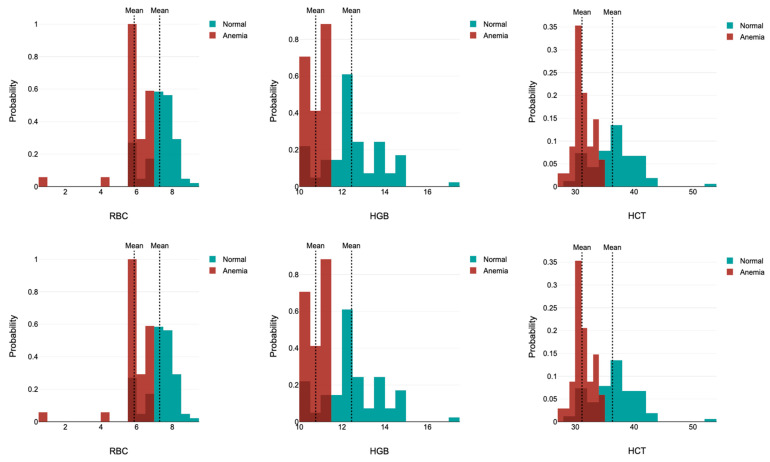
Probability density histograms of the distribution of red blood cell parameters by age group in children with (green bars) and without (red bars) anemia. The figure panels of the 3–5-year-old children are shown above, and the 5–7-year-old children below. The fact that the thresholds for anemia are different for different age groups has caused the distributions of anemia and non-anemia to overlap. Abbreviations: RBC: Red blood cell count, HGB: hemoglobin concentration, HCT: hematocrit.

**Table 1 ijerph-22-01637-t001:** Baseline data of study participants [n = 156].

Variable	Categories	Frequency	%
Sex	Male	74	47.4
	Female	82	52.6
Age (year)	3	40	25.6
	4	34	21.8
	5	30	19.2
	6	22	14.1
	7	30	19.2
District	Chiara	34	21.8
	Huayana	30	19.2
	Huancarama	27	17.3
	Talavera	24	15.4
	Huancaray	19	12.2
	Andarapa	9	5.8
	San Jerónimo	7	4.5
	Otros *	6	3.8

* Includes the districts of Ullputo, Chisi, Chapimarca, Huicihua.

**Table 2 ijerph-22-01637-t002:** Distribution of hematologic parameters by sex. Data in mean ± SD (percentile range 2.5–97.5), *p* < 0.05 (significant).

Hematological Parameters	Total	Male (n = 74)	Female (n = 82)	*p*-Value
WBC (×10^3^/µL)	10.42 ± 1.76 (6.70 to 14.70)	10.47 ± 1.58 (6.60 to 14.29)	10.39 ± 1.87 (6.70 to 14.70)	0.770
LYM (×10^3^/µL)	4.05 ± 1.44 (1.70 to 6.83)	4.06 ± 1.46 (1.74 to 6.90)	4.04 ± 1.43 (1.70 to 6.85)	0.934
MID (×10^3^/µL)	0.78 ± 0.33 (0.30 to 1.60)	0.75 ± 0.29 (0.30 to 1.23)	0.81 ± 0.35 (0.30 to 1.60)	0.276
GRA (×10^3^/µL)	5.57 ± 1.52 (2.80 to 9.03)	5.64 ± 1.54 (3.54 to 9.92)	5.54 ± 1.51 (2.60 to 8.87)	0.684
LYM%	38.50 ± 11.02 (19.30 to 56.00)	38.27 ± 11.49 (19.03 to 56.80)	38.64 ± 10.77 (19.26 to 56.05)	0.836
MID%	7.34 ± 2.87 (2.80 to 12.84)	7.12 ± 2.48 (2.84 to 11.50)	7.48 ± 3.10 (0.30 to 14.07)	0.445
GRA%	54.06 ± 12.60 (33.38 to 76.90)	54.61 ± 12.76 (33.64 to 77.05)	53.71 ± 12.55 (32.00 to 76.90)	0.664
RBC (×10^6^/µL)	6.98 ± 1.08 (4.85 to 8.56)	7.06 ± 0.89 (5.58 to 8.69)	6.94 ± 1.18 (4.14 to 8.56)	0.502
HGB (g/dL)	12.32 ± 2.01 (9.17 to 17.20)	12.54 ± 2.27 (10.14 to 22)	12.18 ± 1.82 (9.08 to 14.93)	0.283
HCT (%)	35.35 ± 4.71 (25.71 to 43.00)	35.52 ± 4.63 (29.19 to 44.70)	35.24 ± 4.79 (25.48 to 43.00)	0.717
MCV (fL)	50.50 ± 2.85 (45.40 to 55.10)	50.44 ± 2.63 (45.17 to 55.37)	50.54 ± 2.99 (45.51 to 55.24)	0.818
MCH (g/dL)	17.33 ± 1.06 (15.90 to 18.60)	17.30 ± 0.81 (15.90 to 18.79)	17.34 ± 1.20 (15.80 to 18.70)	0.817
MCHC (pg/dL)	34.31 ± 1.11 (32.71 to 36.73)	34.32 ± 0.99 (32.56 to 37.20)	34.31 ± 1.18 (32.69 to 36.70)	0.974
PLT (×10^3^/µL)	226.47 ± 35.10 (167.00 to 295.06)	224.50 ± 27.86 (167.00 to 284.36)	227.70 ± 39.04 (135.94 to 311.93)	0.581

Abbreviations: WBC: white blood cell count, GRA% and GRA#: granulocyte count, LYM% and LYM#: lymphocyte count, MID% and MID#: mid-sized white cells, RBC: Red blood cell count, HGB: hemoglobin concentration, HCT: hematocrit, MCV: mean corpuscular volume, MCH: mean corpuscular hemoglobin, MCHC: mean corpuscular hemoglobin concentration, PLT: platelet count.

**Table 3 ijerph-22-01637-t003:** Distribution of hematologic parameters by age group. Data in mean ± SD (percentile range 2.5–97.5), *p* < 0.05 (significant).

Hematological Parameters	Age Group		*p*-Value
	3–4 (n = 74)	5–7 (n = 84)	
WBC (×10^3^/µL)	10.28 ± 1.74 (6.65 to 14.55)	10.55 ± 1.78 (7 to 14.7)	0.350
LYM (×10^3^/µL)	3.94 ± 1.45 (1.79 to 6.8)	4.15 ± 1.43 (1.83 to 7.16)	0.362
MID (×10^3^/µL)	0.73 ± 0.31 (0.30 to 1.6)	0.83 ± 0.34 (0.30 to 1.6)	0.053
GRA (×10^3^/µL)	5.59 ± 1.43 (2.60 to 9.12)	5.56 ± 1.61 (2.8 to 9.21)	0.906
LYM%	37.86 ± 11.14 (18.7 to 55.77)	39.1 ± 10.94 (19.89 to 57.33)	0.494
MID%	6.79 ± 2.72 (0.30 to 11.5)	7.85 ± 2.93 (2.93 to 14.87)	0.021
GRA%	55.14 ± 12.56 (34.2 to 77.58)	53.08 ± 12.63 (32 to 75.91)	0.310
RBC (×10^6^/µL)	7.06 ± 0.96 (4.98 to 8.56)	6.91 ± 1.18 (3.18 to 8.52)	0.369
HGB (g/dL)	12.61 ± 2.48 (8.9 to 22)	12.06 ± 1.42 (10.23 to 14.7)	0.085
HCT (%)	35.64 ± 5 (25.35 to 43)	35.09 ± 4.46 (28.09 to 42.75)	0.474
MCV (fL)	50.51 ± 2.43 (45.4 to 55.1)	50.5 ± 3.19 (44.41 to 55.92)	0.975
MCH (g/dL)	17.26 ± 0.70 (15.9 to 18.5)	17.38 ± 1.31 (15.8 to 19.7)	0.467
MCHC (pg/dL)	34.19 ± 0.92 (32.63 to 36.31)	34.42 ± 1.25 (32.63 to 37.76)	0.198
PLT (×10^3^/µL)	226.61 ± 31 (151 to 301.5)	226.34 ± 38.6 (156.46 to 333.49)	0.962

Abbreviations: WBC: white blood cell count, GRA% and GRA#: granulocyte count, LYM% and LYM#: lymphocyte count, MID% and MID#: mid-sized white cells, RBC: Red blood cell count, HGB: hemoglobin concentration, HCT: hematocrit, MCV: mean corpuscular volume, MCH: mean corpuscular hemoglobin, MCHC: mean corpuscular hemoglobin concentration, PLT: platelet count.

**Table 4 ijerph-22-01637-t004:** Distribution of hematologic parameters according to anemia status in Peruvian children. Data in mean ± SD (percentile range 2.5–97.5).

Hematological Parameters	3–4 Years (n = 74)			5–7 Years (n = 82)		
	Anemia (n = 14)	No Anemia (n = 60)	*p*-Value	Anemia (n = 34)	No Anemia (n = 48)	*p*-Value
RBC (×10^6^/µL)	5.56 ± 0.47 (4.91 to 6.52)	7.06 ± 0.96 (4.98 to 8.56)	<0.00001	5.87 ± 1.02 (4.46 to 6.62)	7.36 ± 0.79 (5.58 to 8.54)	<0.00001
HCT (%)	33.97 ± 6.89 (25.3 to 39.8)	35.64 ± 5 (25.43 to 43)	<0.00001	31.13 ± 1.73 (27.86 to 34.45)	36.54 ± 4 (29.74 to 42.89)	<0.00001
MCV (fL)	49.95 ± 6.43 (46.35 to 53.8)	50.51 ± 2.43 (45.4 to 55.1)	0.223	51.83 ± 3.86 (42.91 to 55.1)	49.75 ± 2.49 (45.54 to 55.1)	0.0018
MCH (g/dL)	17.12 ± 2.15 (15.93 to 18.5)	17.26 ± 0.70 (15.9 to 18.5)	0.001	17.98 ± 1.75 (16.06 to 19.88)	17.08 ± 0.77 (15.8 to 18.58)	0.0004
MCHC (pg/dL)	33.84 ± 4.11 (33.59 to 35.6)	34.19 ± 0.92 (32.6 to 36.31)	0.020	34.64 ± 1.54 (32.94 to 37.79)	34.35 ± 0.99 (32.6 to 36.54)	0.165

## Data Availability

All other data generated and analyzed during the current study are available in this published article. Further inquiries can be directed to the corresponding author.

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
