# Peer review of "Distribution of Hematologic Parameters of Complete Blood Count in Anemic and Nonanemic Children in a Mining-Exposed Highland Peruvian Community"

_ijerph, 2025, doi:10.3390/ijerph22111637_

Round 1

Reviewer 1 Report (Previous Reviewer 1)

Comments and Suggestions for Authors

This is the second revision of this manuscript.  The authors have made some albeit minimal modifications, which have made some improvements, but they have not addressed some major concerns satisfactorily.

1. The authors insistently continue to refer to a necessity to establish 'reference intervals' for children in the mining communities (lines 33, 73, 266, 293, 325, 326).  By definition, "reference intervals" relate only to healthy populations, and the term must NOT be used to relate to or include any population with diseased individuals.  Despite this, there are mentions, for example, of "the necessity of establishing distinct reference intervals for anemic and non-anemic children in mining contexts" (lines 265-6).

It has been stated (lines 32-5) that that there is a need for such "reference intervals in mining communities to facilitate early diagnosis and intervention".  Why can't the international or national reference intervals be used to diagnose anaemia and enable intervention?

It has been stated (lines 73-5) that reference values are needed because "conventional hematological parameters fail to capture the distinct physiological adaptations" of the communities burdened by pollutants.  It is unclear what "conventional hematological parameters" mean.  Also, is there really any physiological adaptation, i.e. changes in physiology to mitigate effects of toxins?  If so, how do the findings of this work capture the distinct physiological adaptations and toxicological burdens in mining communities?  Please either remove this sentence or clarify what you mean, outlining clearly the manner in which your findings can be linked to the outcomes that you envisage.

2.  A crucially important question is 'what percentage of children living in polluted mining communities are anaemic according to an agreed definition of anaemia, and how severe the degree of anaemia is in those affected?  This evidence would enable allocation of public resources to address the issue.  The authors have data on these, and have presented them in Table 4 (but without categorisation according to the severity of anaemia) and also in some fashion in Figure 2.  But they have not presented this important information in the text of the Results section, nor have they made any mention of it in their Discussions.

3.  In the Conclusions section (lines 323-8),

I) It is haemoglobin concentrations that define anaemia, and not that "anaemia status significantly predicts haemoglobin".

II) As questioned above, how do the "intervals established in this research may be useful for evaluating anaemia in children aged 3-7)"?  Assessing anaemia may already be carried out simply by using the definition of anaemia and the haemoglobin thresholds.  The authors themselves have used the Peruvian MINSA's definition for establishing anaemia in their cohort.

III) The final sentence of the Conclusions also needs to be modified according to the points raised above.

4.  My previous point about Figure 2 has not been addressed at all.

I) If the figure is a probability distribution as it appears to be, then the y-axes are incorrect.  The cumulative probabilities should total 1.0, but possibly with the exception of HCT, the cumulative probabilities for RBC and HGB far exceed 1.0.  In fact, one bar for RBC has a probability of 1.0 on its own.

II) What is the difference between the top and bottom panels?  They seem identical.

III) The fact that the thresholds for anaemia are different for different age groups has made anaemic and non-anaemic distributions overlap one another.  Please outline this in the figure legend so the reader has an explanation for the overlap.

5. Please correct the sentences in lines 171, 172-3, 174, 206, 223, etc.

6. Is there any reason why the plots in Figure 1 are in two different colours.

Comments on the Quality of English Language

The quality of English is generally fine, with only some very minor amendments necessary.

Author Response

This is the second revision of this manuscript.  The authors have made some albeit minimal modifications, which have made some improvements, but they have not addressed some major concerns satisfactorily.

  1. The authors insistently continue to refer to a necessity to establish 'reference intervals' for children in the mining communities (lines 33, 73, 266, 293, 325, 326).  By definition, "reference intervals" relate only to healthy populations, and the term must NOTbe used to relate to or include any population with diseased individuals.  Despite this, there are mentions, for example, of "the necessity of establishing distinct reference intervals for anemic and non-anemic children in mining contexts" (lines 265-6).

It has been stated (lines 32-5) that that there is a need for such "reference intervals in mining communities to facilitate early diagnosis and intervention".  Why can't the international or national reference intervals be used to diagnose anaemia and enable intervention?

It has been stated (lines 73-5) that reference values are needed because "conventional hematological parameters fail to capture the distinct physiological adaptations" of the communities burdened by pollutants.  It is unclear what "conventional hematological parameters" mean.  Also, is there really any physiological adaptation, i.e. changes in physiology to mitigate effects of toxins?  If so, how do the findings of this work capture the distinct physiological adaptations and toxicological burdens in mining communities?  Please either remove this sentence or clarify what you mean, outlining clearly the manner in which your findings can be linked to the outcomes that you envisage.

 RESPONSE: Yes, we mention those reference intervals as part of the context; at no point do we mention that we will measure them. This is because the future direction of this study is to establish these parameters based on our distribution results. You ask: Why can't the international or national reference intervals be used to diagnose anemia and enable intervention? And the answer to this question is that while there are international standards for identifying anemia, there may be variations in self-tests, rapid POCT systems for hemoglobin determination, manual hemoglobin identification, etc. In this sense, the ranges can be underestimated if there is no established normal range, because in the Peruvian mining context, there is a deficiency in the determination of hematological parameters and access to automated technologies. On the other hand, in mining communities, we do not determine the different physiological adaptations and toxicological loads because there are no tools to evaluate these contaminants. Likewise, physiological adaptations have not been evaluated because this study focuses on hematology, which is how the study was initially designed.

  1. A crucially important question is 'what percentage of children living in polluted mining communities are anaemic according to an agreed definition of anaemia, and how severe the degree of anaemia is in those affected?  This evidence would enable allocation of public resources to address the issue.  The authors have data on these, and have presented them in Table 4 (but without categorisation according to the severity of anaemia) and also in some fashion in Figure 2.  But they have not presented this important information in the text of the Results section, nor have they made any mention of it in their Discussions.

RESPONSE: We believe this question is equally important: What percentage of children living in polluted mining communities are anemic according to an agreed definition of anemia, and how severe is the degree of anemia in those affected? However, this mining area does not have data on the population with anemia, either in adults or children, that would be useful to us. We evaluated a cohort of patients, but it is possible that a group of children were not included because their participation was voluntary. It would be important to know the total number of childhood anemias in this population, but the focus of the study was not to evaluate this at the population level.

  1. In the Conclusions section (lines 323-8),
  2. I) It is haemoglobin concentrations that define anaemia, and not that "anaemia status significantly predicts haemoglobin".
  3. II) As questioned above, how do the "intervals established in this research may be useful for evaluating anaemia in children aged 3-7)"?  Assessing anaemia may already be carried out simply by using the definition of anaemia and the haemoglobin thresholds.  The authors themselves have used the Peruvian MINSA's definition for establishing anaemia in their cohort.

III) The final sentence of the Conclusions also needs to be modified according to the points raised above.

RESPONSE: The conclusion section has been restructured

  1. My previous point about Figure 2 has not been addressed at all.
  2. I) If the figure is a probability distribution as it appears to be, then the y-axes are incorrect.  The cumulative probabilities should total 1.0, but possibly with the exception of HCT, the cumulative probabilities for RBC and HGB far exceed 1.0.  In fact, one bar for RBC has a probability of 1.0 on its own.
  3. II) What is the difference between the top and bottom panels?  They seem identical.

III) The fact that the thresholds for anaemia are different for different age groups has made anaemic and non-anaemic distributions overlap one another.  Please outline this in the figure legend so the reader has an explanation for the overlap.

RESPONSE: It has been corrected as suggested.

  1. Please correct the sentences in lines 171, 172-3, 174, 206, 223, etc.

RESPONSE: It has been reviewed and corrected.

  1. Is there any reason why the plots in Figure 1 are in two different colours.

RESPONSE: There is no reason why the graphs in Figure 1 should have two different colors. They are only placed this way for a proper presentation of Figure 1, which is why they do not have color legends.

Reviewer 2 Report (Previous Reviewer 2)

Comments and Suggestions for Authors

The previous comments were not systematically addressed and many remain unanswered.

methods

line 100 why were hospitalised children excluded? 

line 207 - what two groups?  please define them

table 4 5-7 years (n = 84) anemia (n =34) no anemia (n=48).  34+48 does not equal 84...is there an error?

Author Response

methods

line 100 why were hospitalised children excluded? 

RESPONSE: This information has been included.

line 207 - what two groups?  please define them

RESPONSE: This information has been included.

table 4 5-7 years (n = 84) anemia (n =34) no anemia (n=48).  34+48 does not equal 84...is there an error?

RESPONSE: It has been corrected

Reviewer 3 Report (Previous Reviewer 3)

Comments and Suggestions for Authors

The manuscript is improved and may be accepted for publication

Author Response

COMMENT: The manuscript is improved and may be accepted for publication

RESPONSE: Thank you for your approval.

Reviewer 4 Report (Previous Reviewer 4)

Comments and Suggestions for Authors

The authors have responded to comments, describing the lack of available data to help determine cause of anemia, making the novel story more complete.

Author Response

COMMENT: The authors have responded to comments, describing the lack of available data to help determine cause of anemia, making the novel story more complete.

RESPONSE: Thank you for your approval.

Round 2

Reviewer 1 Report (Previous Reviewer 1)

Comments and Suggestions for Authors

This is the third amended version of this manuscript that has come for review.

The link between pollution and deranged haematological indices has been known for a very long time. The data obtained in the research outlined in this manuscript could have been used to offer useful information on the prevalence of anaemia in children living in mining communities in highland Peru. However, the outlined aim of the study, the details of some of the presented data, the manner in which the authors claim that the data may be used, and some of the conclusions that have been made are inappropriate and in some instances unscientific. Also, some of the findings have not been interpreted and discussed, I had described much of this together with my concerns in my previous reviews, but the authors have either ignored them or not made any attempt to address them. The responses that they have given have not addressed the issues raised, and the alterations have been only very superficial. I will touch upon a handful of issues (again).

In the aims of the study, the authors say "This research is grounded in the imperative to establish specific diagnostic parameters for vulnerable populations." But, as long back as decades ago, 'specific [anaemia and blood cell] diagnostic parameters' were established, and they are the ones that the authors have themselves measured in their research.

The authors insist on using the term reference intervals and mention the employment of the CLSI guidelines for establishing them in the Methods. I encourage the authors to look into the definition of 'reference intervals' as used in laboratory medicine worldwide. Reference intervals relate ONLY to healthy individuals, and are used in order for assay results in individuals suspected of having a clinical abnormality to be assessed against them. Reference intervals must NOT include assay results from diseased or unhealthy individuals, or even individuals that are known to be at risk of having a related condition that can affect the results. Indeed, reference intervals should be established for local populations, but the recruitment of blood donors should be only from those that are known NOT to have any clinical condition that may be affecting the test results. It is simply wrong to use the term 'reference interval' based on the test results of a population that is so clearly impacted by environmental toxins and/or poor nutrition, and whose laboratory results harbour a bias. In fact, the authors themselves mention that "Anemia imposes a disproportionately high burden here, affecting 61.9% of children, a rate escalating markedly over the past decade".

The authors state that reference values are needed because "conventional hematological parameters fail to capture the distinct physiological adaptations" of the communities burdened by pollutants.". This does not make any sense. The parameters are capturing the pathology, and physiological adaptations cannot compensate for the pathology in the studied group..

Some of the data presented appear to be incorrect. In Table 3, the MCV values have 2.5-97.5 centile ranges that span 44.41-55.92 fL, which is clearly far too low to be believable, since healthy children of those age groups have MCVs with lower limits of reference intervals in mid-70s fL (at 2 standard deviations below mean). In the same Table, the MCH values are also similarly too low (2.5-97.5 centile ranges that span 15.9 to 19.7 g/dL, but the lower reference intervals in children of those age groups are around 24 g/dL).

In Table 4, the 2.5-97.5 centile range for haematocrit in the anaemic group doesn't even include the mean value.

Comparisons between white cells have been made between the two age groups, but nowhere in either the Results or the Discussion has it been mentioned as to whether those results were normal or abnormal. In the Discussions, it has been said that "our inclusion of WBC and platelet counts enabled identification of additional heavy metal-induced hematopoietic effects beyond anemia", but the authors don't say what effects were identified. In the Discussions, the authors say that their "inclusion of WBC and platelet analyses .... uncovered potential hematopoietic disruptions", but there is no description or outline of any haematopoietic disruption related to white cells or platelets in their text.

The prevalence of anaemia, which is very important, has not been mentioned, described or discussed in any text, neither in the Results nor in the Discussions. The only reference to it is in Table 4, only where the numbers of anaemic and non-anaemic children have been stated.

The axes of Figure 2 are still incorrect. The cumulative probabilities should total 1.0, but the cumulative probabilities for RBC and HGB far exceed 1.0. As it stands, for example for anaemic children the probability that their RBC values are between 5.5 and 6.0 is 1.0, i.e. all anaemic children's RBC values should be 5.5-6.0. But some anaemic children have other RBC values, which means that the cumulative probability exceeds 1.0 for that group.

The authors give a summary of their findings in the first paragraph of the Discussion by saying that their "results demonstrated that, while most distribution of hematological parameters were similar across sexes and age groups, children tested at all ages with anemia exhibited significantly lower RBC, HGB levels, and HCT values.". They reiterate this in the Conclusions as their main finding. But isn't a significantly lower RBC, Hb and Hct expected in anaemia anyway? In fact, isn't it a low Hb (usually with concomitantly decreased Hct and RBC) that defines anaemia?

The authors also mention that their research addresses "a critical gap in laboratory medicine for clinically interpreting hematological data in high-exposure settings.". Is there really any gap or difficulty in "interpreting" haematological data in the studied community? The authors themselves had no difficulty interpreting the children's data, and were able to say which children had anaemia and which ones did not.

Under their 'Main findings' title, the authors say that their "study can help assess red blood cell abnormalities in Peruvian children residing in mountainous mining regions", but it is unclear how their data helps with this. Most of what has been mentioned under the 'Main findings' title, does not relate to the findings of their research. In short, the Discussion section does not discuss in any useful detail what the data showed and what the research found.

Author Response

In the aims of the study, the authors say "This research is grounded in the imperative to establish specific diagnostic parameters for vulnerable populations." But, as long back as decades ago, 'specific [anaemia and blood cell] diagnostic parameters' were established, and they are the ones that the authors have themselves measured in their research.

The authors insist on using the term reference intervals and mention the employment of the CLSI guidelines for establishing them in the Methods. I encourage the authors to look into the definition of 'reference intervals' as used in laboratory medicine worldwide. Reference intervals relate ONLY to healthy individuals, and are used in order for assay results in individuals suspected of having a clinical abnormality to be assessed against them. Reference intervals must NOT include assay results from diseased or unhealthy individuals, or even individuals that are known to be at risk of having a related condition that can affect the results. Indeed, reference intervals should be established for local populations, but the recruitment of blood donors should be only from those that are known NOT to have any clinical condition that may be affecting the test results. It is simply wrong to use the term 'reference interval' based on the test results of a population that is so clearly impacted by environmental toxins and/or poor nutrition, and whose laboratory results harbour a bias. In fact, the authors themselves mention that "Anemia imposes a disproportionately high burden here, affecting 61.9% of children, a rate escalating markedly over the past decade". The authors state that reference values are needed because "conventional hematological parameters fail to capture the distinct physiological adaptations" of the communities burdened by pollutants.". This does not make any sense. The parameters are capturing the pathology, and physiological adaptations cannot compensate for the pathology in the studied group.

RESPONSE: Dear reviewer, your suggestion has been considered. Any definition or description of reference intervals has been reviewed and removed, as the objective of the study is to describe the variation in hematological parameters in the study population. If at any point we considered mentioning the importance of reference intervals, they have been eliminated. Furthermore, only in the limitations section is it mentioned that these parameters could be evaluated in the future in healthy communities in the mining setting. Please review the text.

Some of the data presented appear to be incorrect. In Table 3, the MCV values have 2.5-97.5 centile ranges that span 44.41-55.92 fL, which is clearly far too low to be believable, since healthy children of those age groups have MCVs with lower limits of reference intervals in mid-70s fL (at 2 standard deviations below mean). In the same Table, the MCH values are also similarly too low (2.5-97.5 centile ranges that span 15.9 to 19.7 g/dL, but the lower reference intervals in children of those age groups are around 24 g/dL).

In Table 4, the 2.5-97.5 centile range for haematocrit in the anaemic group doesn't even include the mean value.

RESPONSE: We have reviewed Table 3 and the raw data, the reported values ​​are correct.

Comparisons between white cells have been made between the two age groups, but nowhere in either the Results or the Discussion has it been mentioned as to whether those results were normal or abnormal. In the Discussions, it has been said that "our inclusion of WBC and platelet counts enabled identification of additional heavy metal-induced hematopoietic effects beyond anemia", but the authors don't say what effects were identified. In the Discussions, the authors say that their "inclusion of WBC and platelet analyses .... uncovered potential hematopoietic disruptions", but there is no description or outline of any haematopoietic disruption related to white cells or platelets in their text.

RESPONSE: Although the study's objective was not to analyze leukocyte parameters, we have included a description of the results and a brief discussion of the most important aspects. It certainly merits further research. Thank you for the suggestions.

The prevalence of anaemia, which is very important, has not been mentioned, described or discussed in any text, neither in the Results nor in the Discussions. The only reference to it is in Table 4, only where the numbers of anaemic and non-anaemic children have been stated.

RESPONSE: Although it is not the objective of the study to find the prevalence, it has been mentioned at the beginning of the presentation of results.

The axes of Figure 2 are still incorrect. The cumulative probabilities should total 1.0, but the cumulative probabilities for RBC and HGB far exceed 1.0. As it stands, for example for anaemic children the probability that their RBC values are between 5.5 and 6.0 is 1.0, i.e. all anaemic children's RBC values should be 5.5-6.0. But some anaemic children have other RBC values, which means that the cumulative probability exceeds 1.0 for that group.

RESPONSE: These graphs are probability density histograms. As previously explained, they show the probability density of the data for each parameter. Please consider the proposed values, as they have also been reviewed and satisfactorily validated by other reviewers of the article.

The authors give a summary of their findings in the first paragraph of the Discussion by saying that their "results demonstrated that, while most distribution of hematological parameters were similar across sexes and age groups, children tested at all ages with anemia exhibited significantly lower RBC, HGB levels, and HCT values.". They reiterate this in the Conclusions as their main finding. But isn't a significantly lower RBC, Hb and Hct expected in anaemia anyway? In fact, isn't it a low Hb (usually with concomitantly decreased Hct and RBC) that defines anaemia?

RESPONSE: Yes, a reduction in these values ​​is expected; however, these changes have not been reported among the child population in Andahuaylas. In this sense, our results confirm what was previously expected. We've adjusted the text, as low RBC and HCT levels are expected. We also mentioned the prevalence of anemia in the study population.

The authors also mention that their research addresses "a critical gap in laboratory medicine for clinically interpreting hematological data in high-exposure settings.". Is there really any gap or difficulty in "interpreting" haematological data in the studied community? The authors themselves had no difficulty interpreting the children's data, and were able to say which children had anaemia and which ones did not.

RESPONSE: We have reviewed this paragraph and modified it according to the suggestion made.

Under their 'Main findings' title, the authors say that their "study can help assess red blood cell abnormalities in Peruvian children residing in mountainous mining regions", but it is unclear how their data helps with this. Most of what has been mentioned under the 'Main findings' title, does not relate to the findings of their research. In short, the Discussion section does not discuss in any useful detail what the data showed and what the research found.

RESPONSE: We have adjusted the text according to the reviewer's suggestions.

This manuscript is a resubmission of an earlier submission. The following is a list of the peer review reports and author responses from that submission.

Round 1

Reviewer 1 Report

Comments and Suggestions for Authors

The authors have addressed some of the issues that were previously raised, and have made some changes and adjustments. These have improved in particular the presentation of data. However, unfortunately, the main criticisms of the paper, namely the focus and the message of the paper, have not been addressed satisfactorily.  There has been some rewording of some sentences, but the focus of the paper on reference intervals and the diagnosis of anaemia has not changed.

In the Abstract, the authors write that their findings "highlight the need for personalised hematological reference intervals in mining-exposed populations, which may facilitate early diagnosis and intervention for anaemia". In the Conclusions, they state again that "the intervals established in this research may be useful for the evaluation of anemia in children aged 2-7 years". In the Discussions, they say that "reference intervals in high-exposure settings are critical to enhance diagnostic accuracy". It is unclear why the internationally-agreed WHO criteria and reference intervals could not be used for diagnosing anaemia and making interventions.

Again, in lines 73-76, the authors say that "current literature underscores the imperative for population-specific reference values" to "capture the distinct physiological adaptations and toxicological burden of Andean highland communities exposed to mining pollutants". However, the basic criterion for establishing population-specific reference values dictates that any data obtained from relevant unhealthy individuals should be excluded.  This seems to have been ignored.

The authors report hamoglobin concentrations as low as 8.9 mg/dL in non-anaemic children (line 28). According to WHO (2024), anaemia is defined as a haemoglobin (in g/dL) of <10.5 for children of age 5-23 months, <11.0 for age 24-59 months, <11.5 for age 5-11 years and, <12.0 for those aged 12+ years. On this basis, the authors have not used the correct definition of anaemia. Although they say that anaemia was diagnosed by the criteria of the Ministry of Health (MINSA) of Peru (lines 103 onwards), the data in Figure 1 does not even support that. In figure 1, normal children have Hb concentrations down to 10.0 g/dL, indicating that children with mild anaemia (according to MINSA) were also included in the normal group. The inclusion of children with mild anaemia in the non-anaemic group neither has been mentioned in the Methods nor would be acceptable.

Furthermore, WHO also advises on adjusting haemoglobin reference interval according to elevation above sea level, which would be required for the mountainous regions where the study was carried out. This has not been taken into account in the study. As an example, an elevation of 2,500 meters above sea level would mean that the haemoglobin threshold for anaemia would need to be increased from 11.0 g/dL to 12.1 g/dL.

A few relatively smaller issues exist:

The data in reference 21 indicated that in one of the three regions where mining occurred and were studied, 54.6% of children suffered from malnutrition and 71.5% suffered from anaemia. There were similarly alarmingly high rates of malnutrition and anaemia in the other two regions as well. Therefore, a significant contributor to anaemia in the studied population may be the prevalent malnutrition, but virtually no attention to that has been paid in this study.

In lines 298-301, the authors mention that they "did not analyze the frequency or type of exposure to mining toxins or the concentration of common toxins, so the distribution of the parameters could be bimodal". I cannot see how the type and amount of exposure to toxins could make the distribution of the findings bimodal when they have had and the of data, and its analyses have not shown any bimodal distribution.

The y-axes in Figure 2 should be frequency or number of chuldren; not probability.

An agreed style for quoting references should be employed. Most journals have been abbreviated properly, but some have their full names. The style of others need to be amended, e.g. references 22 and 23.

There are a few typographical errors.

Author Response

Kindly check the attached document.

Reviewer 2 Report

Comments and Suggestions for Authors

Most comments have been addressed.  A major limitation is comparability to a 'normal' population within Peru who have not been exposed to mines.  This should be stated within the discussion.  The use of distribution profile is much more accurate than a reference range and the manuscript is now more appropriately scoped for the data presented.

Author Response

Kindly check the attached document.

Reviewer 3 Report

Comments and Suggestions for Authors

The authors have significantly improved their manuscript. However, there are some technical mistakes that need to be corrected: e.g. malr, instead of male. Also, some text in Spanish is left in the manuscript as: ELIANE citar la referencia 1 de arriba la de MINSA.

Comments on the Quality of English Language

The manuscript is written well, however, there are some tupos that need to be corrected.

Author Response

Kindly check the attached document.

Reviewer 4 Report

Comments and Suggestions for Authors

This manuscript describes an important study of children 3-7 years in Peruvian highlands 156, with nearly half being 3-4 years in mining contaminated areas. Age group was important, before the major sex differences seen and no difference in heme parameters by sex. The authors define the anemic ones were selected for different Hb in 24-59 month-olds was mild at 10-10.9 and moderate 7-9.9; with 5-7 year-olds was mild at 8-10.9 and moderate <8. This is a thoughtful way to display the data, by “anemia” because one could examine other RBC and WBC parameters on the Sysmex semiautomated heme counter.

The first concern of this paper is that the authors do not report which heavy metals contaminate these mining communities, but many divalent heavy metals can compete with iron for sites on the cell surface iron transporter, divalent metal transporter-1 (DMT-1), including mercury, cadmium, manganese, nickel, cobalt and lead. These heavy metals can also cause a relative vitamin B12 and folate deficiency.

Although no difference in total WBC count, there were mild elevation in Mid% in older kids (Mid% is mostly monocytes and eosinophils on Sysmex machines). Without a manual differential by microscope slide exam, this is hard to interpret. This finding is unusual, because there is generally not an expected age difference in monocyte percentages, but that eosinophil percentages are expected as higher, not lower in younger children. It may be possible that vitamin B12 and folate deficiency are seen in these children and that hypergegmented neutrophis may be appear in the Mid% instead of the granulocyte percentage because they physically appear more like circulating monocytes.

As anticipated, the RBC indices differences between anemic and nonanemic were observed, but indices in the anemic children were not all consistent with the microcytic, microchromic anemia of iron deficiency. Microscopic slide exams here would have been important in these children to help determine cause. Assuming that this was not performed, a potential option is to report variability in RBC size. It is important to note that RBC may be in the circulation between 90-120 days, so more than one population of cells could be seen. Normal RBC generally start out large and get smaller over the lifespan. Exceptions include either iron deficiency where cells start out smaller (microcytic) and get smaller yet as they age…or in B12 and Folate deficiency when they start out large and get smaller but stay relatively large (macrocytic). Iron deficiency and B12/folate deficiency may cause a shorter lifespan of these cells.

Most Sysmex machines also report RDW, which is the coefficient of variation of size of RBC. This parameter is very useful when combined with the other RBC indices. RDW can be higher in iron deficiency anemia when combined with lower MCV, and MCH, MCHC.

The authors found that in the anemic younger children MCV was unchanged, but MCH and MCHC were higher. This would be described as a normocytic, normochromic or even hyperchromic anemia. This is not typical of iron deficiency anemia. A larger RDW would be helpful in this case if there was a mixed population of cells of variable size and ages, including smaller cells of iron deficiency. Or if there are smaller (iron deficiency) and larger (B12/folate deficiency) cells

Another finding of interest was that in the anemic older children, MCV and MCH were both higher. Thus, this anemia would be described as a macrocytic, normochromic anemia, more consistent with vitamin B12 and folate deficiency. Thus, RDW would be useful to help determine if the cell population is more homogeneous such as seen in relative macrocytosis of B12 or folate deficiency or a mix of older and younger (larger cells).

If there are no microscopic slides available, even in a subset of children, checking for availability of RDW would improve the paper. If these are not available, at least discussing RBC age, size variation, and potential iron deficiency and B12/folate deficiency in the paper would be helpful.

It may be important to add at least WBC count to Table 4 to address whether there is any evidence for anemia of inflammation, something that may occur in this population.

Some results from easy to understand tables are duplicated in the text, sometimes ranges and p values. If the editors think this takes up too much space, they could be simplified by saying for example, line 182-187, "The WBC and its subsets when expressed by absolute number or percent (table 2) did not differ by sex." 

Author Response

Kindly check the attached document.
